# Lower Jaw Full-Arch Restoration: A Completely Digital Approach to Immediate Load

**DOI:** 10.3390/healthcare12030332

**Published:** 2024-01-28

**Authors:** Claudia Todaro, Michael Cerri, Ruggero Rodriguez y Baena, Saturnino Marco Lupi

**Affiliations:** 1School of Dentistry, Department of Clinical Surgical, Diagnostic and Pediatric Sciences, University of Pavia, 27100 Pavia, Italy; ruggero.rodriguez@unipv.it (R.R.y.B.); saturninomarco.lupi@unipv.it (S.M.L.); 2Private Practice, 29011 Borgonovo Val Tidone, Italy; michael.cerri90@gmail.com

**Keywords:** immediate loading, bone loss, digital planning, Toronto Bridge protocol, surgical guides, 3D printing, full digital workflow, laser sintering, laser melting

## Abstract

The digital transformation has revolutionized various sectors, including dentistry. Dentistry has emerged as a pioneer in embracing digital technologies, leading to advancements in surgical and prosthetic oral healthcare. Immediate loading for full-arch edentulous dental implants, once debated, is now widely accepted. This case report describes a 74-year-old patient with dental mobility and significant bone loss who was rehabilitated using a Toronto Bridge protocol on four dental implants with immediate loading. Digital planning, surgical guides, 3D printing, and precision techniques were employed. The surgery involved implant placement and prosthetic procedures. The patient reported minimal post-operative discomfort, and after four months, the definitive prosthesis was successfully placed. This case demonstrates the efficacy of immediate loading in complex dental scenarios with digital innovation, resulting in improved patient outcomes. The full digital workflow, including 3D printing and the use of modern materials, enhances the efficiency and predictability of oral rehabilitation, marking a transformative era in dental care. The integration of digital technology in all phases of treatment, from diagnosis to finalization, makes this approach safer, reliable, and efficient, thereby benefiting both patients and clinicians.

## 1. Introduction

Over the past two decades, there has been a significant digital transformation that has impacted various sectors, dentistry included [1,2,3].

Dentistry has quickly adopted digital advancements in daily practice, leading to pioneering progress. Research has not only delved into new biomaterials, but has also driven the evolution of implant technologies and sophisticated prosthetic devices, thus significantly transforming the dental care landscape compared to just a few years ago [4]. This digital wave has indelibly transformed every aspect of clinical practice, from the initial stages of diagnosis and meticulous planning of treatment protocols to complex and sensitive procedures conducted in complex clinical and surgical contexts [5,6]. The production of prosthetic devices has undergone a profound transformation due to the pervasive influence of digitalization. This change is observable at every stage and has led to accelerated processes, enhanced precision based on recent studies, and an elevation of the overall standard of patient care. A notable outcome arising from the integration of clinical –biological research and materials engineering is the widespread adoption of immediate loading for full-arch edentulous dental implants [7]. Initially a topic of contention among dental professionals, this practice has undergone rigorous research and extensive clinical studies, solidifying its credibility and legitimacy in dental practice through medium- and long-term follow-up investigations [8,9,10]. The research on this topic has not only affirmed the legitimacy of immediate loading in dental practice, but has also underscored its clinical significance, boosting confidence among both dental professionals and patients [8]. The combination of immediate loading with personalized treatment planning through computer-guided surgical prosthetic protocols has revolutionized the field. This innovative approach provides various clinical advantages, with a notable reduction in treatment times being a key benefit, facilitated by the possibility of pre-printing prosthetic products before surgery. Patients quickly recognize and appreciate the streamlined and efficient processes, enhancing their overall dental care experience [11]. The scientific literature serves as a repository of studies strongly supporting this protocol, particularly in cases of complete mandibular edentulism involving either overdentures or fixed prostheses. Consequently, a comprehensive digital protocol opens intriguing possibilities for addressing complex cases with immediate loading. The presented case study vividly illustrates the latest methods in managing intricate dental scenarios, emphasizing prosthetic design and the use of modern manufacturing practices. These techniques, characterized by precision, attention to detail, and pre-production capabilities, have progressively gained validation within the scientific community [5,12,13,14,15].

Laser sintering and laser melting are advanced production techniques utilizing laser technology in dentistry used to create dental and prosthetic restorations through additive manufacturing (AM) or 3D printing. These processes, such as selective laser sintering (SLS) and selective laser melting (SLM), construct customized components layer by layer based on digital models. Despite traditional subtractive methods prevailing in dental practice, there is a growing interest in additive manufacturing. These technologies, adaptable to various dental materials, stand out in prosthetic dentistry for their precision and versatility. In summary, laser sintering and laser melting offer advantages over traditional methods, providing improved mechanical properties, accuracy, efficiency, and material versatility in dental applications [16,17].

This clinical case serves as a captivating exploration of the latest digital methodologies in achieving a comprehensive rehabilitation, making it particularly intriguing. The case is distinctive for its thorough examination of cutting-edge digital techniques applied to address a complex scenario. It is a source of valuable insights as it navigates through a series of substantial limitations, thereby influencing specific therapeutic choices meticulously considered during the digital design phase. In essence, this case study symbolizes the intersection of technology and dental practice, shedding light on the potential to elevate patient outcomes through the transformative capabilities of digital innovation. Efficiency, precision, and advanced practices promise a more predictable and successful future for oral rehabilitation procedures.

## 2. Materials and Methods

A 74-year-old female patient required dental rehabilitation treatment in the lower jaw, mainly due to advanced tooth mobility in the frontal sector and peri-implantitis to the implants in the diatoric regions. Her medical history demonstrated excellent overall health and she was classified as ASA I, following risk assessment by the American Society of Anesthesiologists [18].

The clinical examination highlighted severe mobility on all residual elements assessable as grade II and grade III according to the Miller scale [19] (Figure 1).

The CT scan showed the expected bone reabsorption in the frontal region, but also highlighted the failure of the implant in zone 4.4 and consequently the bridge in quadrant IV. The residual implant in the quadrant III also showed signs of failure (Figure 2).

Overall, the diatoric regions of quadrants III and IV presented virtually no remaining alveolar bone.

The patient refused any kind of removable denture, even one supported by implants. GBR techniques were refused as well as any treatment for the upper arch. Considering the limited amount of bone available, the expected bone reabsorption, and the patient’s need for an immediate load, the All-on-Four Computer-Guided protocol was chosen for the rehabilitation. After submitting and obtaining the informed and written consent, surgical and prosthetic planning was carried out as illustrated below.

### 2.1. Planning

#### Surgical Guide

After the patient accepted the treatment plan, intraoral scans were taken and it was decided to keep the DVO unchanged for at least the provisional prosthesis.

Through Digital Smile Design (DSD) the wax-up was created with Exocad software (Galway 3.0, Exocad^®^, Darmstadt, Germany) and imported in the surgical guide Implant3D^®^ software (version 9.2.1, Medialab Spa, Milan, Italy). The implants were positioned between the branches of the alveolar nerves, considering the emergence profiles suggested by the wax-up and residual bone (Figure 3).

Four 3.75 × 13 mm implants (Intra-Lock^®^ System Europa Spa, Salerno, Italy) were planned. The implants in regions 4.4 and 3.4 were tilted by 30° to ensure proper emergence in zones 4.5 and 3.5 of the provisional prosthesis, respectively, while avoiding the foramen. The 4 mm bone deficiencies that were evident on both implants were assessed, but it was not deemed necessary to proceed with Guide Bone Regeneration (GBR) (Figure 4).

An important bone reabsorption was expected after tooth extraction; for this reason, implants 3.2 and 4.2 were positioned below bone level and the sleeves had to be elevated by 2 mm (Figure 5). The immediate post-extraction loading scenario requires a more aggressive implant, generally conical, to obtain sufficient torque [20]. For this reason, the implant to be considered should have been a 4.0 mm Conic CT (Intra-Lock^®^ System Europa Spa, Salerno, Italy). However, based on the bone density and cortical thickness present, a cylindrical-shaped implant was more appropriate. To be able to satisfy all surgical and prosthetic needs, the 3.75 mm conical implant was used, but with a less aggressive spiral compared to the 4 mm Conic CT one [21].

On-Flat^®^ abutments (Intra-Lock^®^ System Europa Spa, Salerno, Italy) were chosen based on the expected amounts of residual bone and gingiva following tooth extraction (Figure 6).

Scan abutments were added and the STL file was exported to Exocad^®^ (Exocad^®^, Darmstadt, Germany) to finalize the provisional denture (Figure 6).

Due to the inadequate supports available for a purely dental supported surgical guide, especially in the frontal region, a hybrid approach was chosen with support pins in addition to residual elements. To place the support pins, a first guide was created sustained by the elements between 3.3 and 4.3 (Figure 7).

The second guide used to place the implants retained the same support pins to guarantee stability in the frontal region and was sustained using 3.3, 4.6, and 3.6 for additional support (Figure 8).

Due to their position, the extractions of the remaining teeth and of fixture 4.4 were scheduled before the insertion of the implants. The only elements to be extracted after insertion of the fixtures, such as support elements for the surgical guide, were 3.3. and 3.6. The implant in region 4.6 was deemed valid and was therefore maintained.

### 2.2. Provisional Prosthesis

Using the Scan Abutment STL, wax-up, and gingiva, the provisional prosthesis was finalized and prepared for 3D printing.

### 2.3. Three-Dimensional Printing

The surgical guides were printed using Keyprint^®^ (Keystone, Singen, Germany). Sleeves for implants and support pins were provided by Medialab (Medialab Spa, Milan, Italy). The temporary prosthesis was 3D printed using NextDent^®^ C&B MFH (NextDent B.V., Soesterberg, The Netherlands). Both the devices were printed on the MoonNight^®^ 3D printer (Vertysystem, Vicenza, Italy).

### 2.4. Surgery

The intake of 2 g of amoxicillin + clavulanic Acid was prescribed one hour before the operation to be continued for five days with 1 g every twelve hours. To shorten the duration of the main operation, the implant in zone 4.4 was extracted the day before.

The following day (right before surgery), the patient was asked to rinse for one minute with 0.30% chlorhexidine to ensure proper oral disinfection. Two vials of Articain (1:100.000 adrenaline) were used during the operation.

The first surgical guide was checked for fitting and the anchor pins were placed (Figure 9).

Once the first guide was removed, the elements 4.2, 4.1, 3.1, 3.2, and 3.3 were extracted and the alveoli were carefully cleaned from residual dental tartar and inflammatory tissue.

After teeth removal and disinfection with 1% chlorhexidine gel, the second guide was fitted with the addition of the anchor pins previously prepared with the first guide (Figure 10).

The Intra-Guide^®^ protocol (Intra-Lock^®^ System Europa Spa, Salerno, Italy) was followed for the guided preparation of the implant sites. Considering the steep shape of the residual bone, particular care was given to the osteotomy to avoid any drift of the subsequent preparation drills (Figure 11).

After the osteotomy, drills of increasing length (6.5 mm, 8 mm, 10 mm, 11.5 mm, and 13 mm) and diameter (2.0 mm and 3.2 mm) were used to create an adequate implant socket. All of the implants achieved an insertion torque between 40 and 45 Nm (Figure 12).

The bone deficiency on implants 3.4 and 4.4 was evaluated and the decision not to use GBR protocols was confirmed (Figure 13).

Residual alveolar bone was smoothed to achieve a flat ridge with the implants 2 mm below the bone upper margin except for the buccal versant of implants 3.4 and 4.4.

Three-millimeter On-Flat^®^ abutments (Intra-Lock^®^ System Europa Spa, Salerno, Italy) were screwed in (Figure 14). Titanium cylinders were screwed into the On-Flat^®^ abutments.

The temporary prosthesis was cemented on titanium cylinders screwed into On-Flat abutments. The patient was discharged after being provided with all the necessary post-operative instructions. The patient reported minimal post-operative swelling, hematoma, and pain. The sutures were removed after 7 days.

### 2.5. Definitive Prosthesis

After 4 months, the healing was proceeding well without major issues. Both CBCT and clinical exam confirmed that all four implants were healed. Ridge reabsorption was measured around 1 to 2 mm and the bone on the buccal side of implants 3.4 and 4.4 was restored as expected. Due to the gingival reabsorption, the On-Flat^®^ abutments on 3.2 and 4.2 were swapped with the 2 mm height version (Figure 15).

Intraoral scans with scan abutments in place were taken and the provisional prosthesis was screwed in again. The scan abutment STL obtained from the digital planning and the one obtained after 4 months were superimposed to check the accuracy of implant insertion (Figure 16).

Another Digital Smile Design and wax-up were performed and the substructure was created using the Exocad Bar module. The DVO remained unchanged (Figure 17).

The substructure was 3D printed in Cr-Co using laser melting (88dent Spa, Pero, Italy) and the teeth were 3D printed using NextDent^®^ C&B MFH (NextDent B.V., Soesterberg, The Netherlands) on a MoonNight^®^ 3D printer (Vertysistem, Vicenza, Italy). The substructure was passivized and cemented on titanium cylinders (Figure 18).

During the same sitting, aesthetics were checked, and occlusion adjustments were carried out. The definitive prosthesis was screwed in during the following sitting (Figure 19).

The one-year follow-up showed excellent stability of the result. Bone level increased in zones 4.4 and 3.4 without recurring GBR, and reabsorption in zones 4.1 and 3.1 stopped at the height predicted during the surgical planification (Figure 20).

## 3. Discussion

Dental implantology has recently undergone a significant and thrilling transformation propelled via technological advancements, propelling the discipline into new fields of research. Three-dimensional printing and associated manufacturing techniques offer fresh opportunities for immediately loaded implant procedures [22] Particularly, laser sintering and laser melting have emerged as cutting-edge technologies. These processes involve additive manufacturing via the utilization of high-intensity lasers to precisely melt and fuse metal powder particles layer by layer. The outcome is the creation of intricately detailed, personalized dental components that exemplify precision and complexity [23,24].

These advanced manufacturing techniques mark a paradigm shift in how prostheses, especially implant-supported prostheses, are managed and designed [25]. While the precision achieved using laser sintering and laser melting is an important feature, their impact goes much further. These techniques also offer the use of a wide range of materials, including biocompatible options such as titanium, cobalt–chromium alloys, and specialized dental alloys [26]. This versatility of the materials guarantees that the components that revolve around the implant-supported prosthesis can be developed with specific characteristics for each individual case to satisfy the specific needs of each patient, considering factors such as biocompatibility and resistance [27]. The ability to precisely select and tailor materials is essential to provide the best possible results and a favorable prognosis in long-term follow-up [28,29,30,31,32]. In addition to the clinical benefits, the efficiency of laser sintering and laser melting in the production of prosthetic elements promises to reduce time and costs [33,34]. This is a win–win situation, as it improves the convenience and accessibility of these advanced techniques for practitioners and translates into more convenient solutions for patients [35]. Digital tools and computerized planning of implant–prosthetic rehabilitation represent the fundamental pillars of modern dentistry. The degree of digitalization shown in this case study not only allows the surgery to be managed and planned with superior precision, but also allows the clinician and the dental technician to cooperate more effectively, thus increasing the chances of a rehabilitation with long-term success [36]. In this particular clinical case, which presented significant challenges to the clinician, a more comprehensive pre-surgical study was deemed necessary. Special efforts were made to implement prosthetic strategies pertaining to the substructure in order to optimize the cleanability of the prosthetic device. Furthermore, the decision to position the anterior implants more lingually than what was prosthetically desired underwent careful examination during both the pre-operative and prosthetic design phases by considering various factors. These factors encompassed the patient’s limitations in undergoing GBR procedures, the deficiency of bone on the vestibular side, and the choice of the patient to not rehabilitate the upper arch. These variables were scrutinized in detail, leveraging the preview capabilities enabled via computer-guided planning. Additionally, from a prosthetic standpoint, precise measurements were employed to thoroughly assess the maximum possible extension of the cantilevers and to structurally align the loads accordingly due to the constrained surgical choices. At the one-year follow-up, it was clear that bone levels had stabilized at the height programmed during the pre-surgical study, a testament of the adequate pre-visualization and execution of the surgery. The implants, originally placed subcrestally during surgery, were comfortably at bone level after 1 year, while the On-Flat abutments elevated the interface with the prosthesis at tissue level, guaranteeing cleanability and protection for the underlying implant. This case study reveals, by means of an analysis of superposition of pre- and post-surgical STL files, the discrepancy in the actual position of the implants compared to the planned one. Measuring placement accuracy, as shown in this work, is simpler to carry out compared to superimposing DICOM files from CBCT scans and saves the patient from additional radiation.

The accuracy varies from 0.1 to 0.4 mm in terms of transversality, and from 0.6 to 0.7 mm in terms of height. Although a statistical analysis was not conducted due to the limited size of the sample, this clinical case suggests avenues for future scientific research that can verify the reliability and accuracy of the various components and techniques involved in such rehabilitations. Despite the limitations linked to the type of study presented, it is believed that the value of this clinical case may be of particular interest for professionals in this sector. Currently, there are no recognized protocols to refer to for the completely digital management of immediate post-extraction rehabilitations of the entire arch, where the surgical component is guided using multiple templates and the prosthetic component is created and produced even before the operation is performed. An objective comparison at each stage of this clinical case with others present in the literature is challenging due to the absence of validated protocols, thus the achievement of comparable therapeutic goals involves following different paths in the design and use of digital tools [37,38]. This study highlights the need for further scientific investigation and development to bridge this gap and provide clear guidelines for clinical practice in similar contexts. Reaffirming the limitations of the study, the presented protocol provides interesting foundations to be extended to a larger sample. From a clinical perspective, although the workflow required a significant amount of design time, it also enabled more efficient surgical procedures and very rapid delivery of the prosthesis. Notably, a positive outcome emerged from the follow-up checks, serving as a reassuring source of long-term stability.

## Figures and Tables

**Figure 1 healthcare-12-00332-f001:**
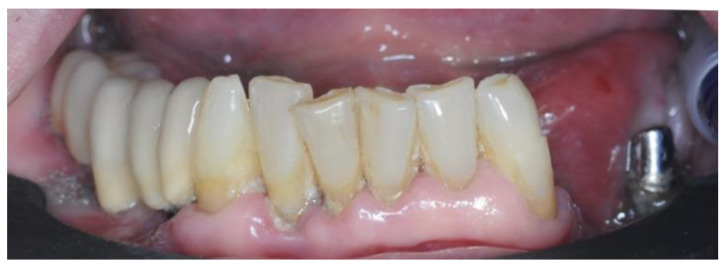
Front view depicting the intraoral state of the lower arch.

**Figure 2 healthcare-12-00332-f002:**
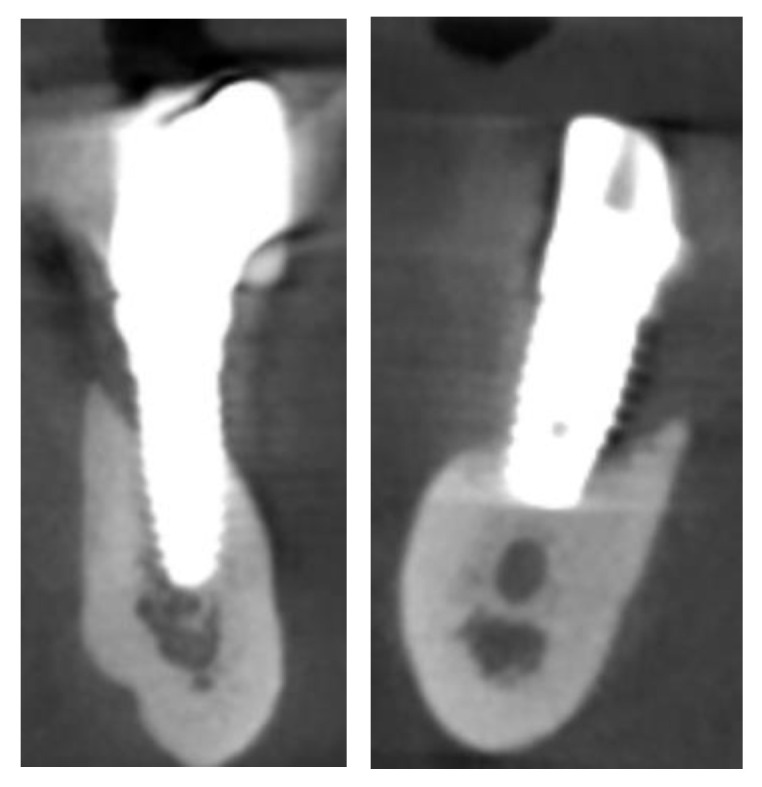
CBCT slices highlighting peri-implantitis on the implants in zones 44 and 36.

**Figure 3 healthcare-12-00332-f003:**
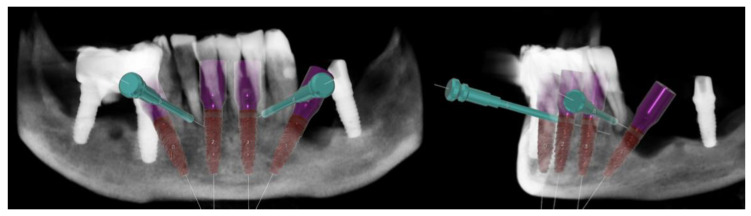
Details of the digital planification of the implant placement surgery with abutments and scan abutments in place.

**Figure 4 healthcare-12-00332-f004:**
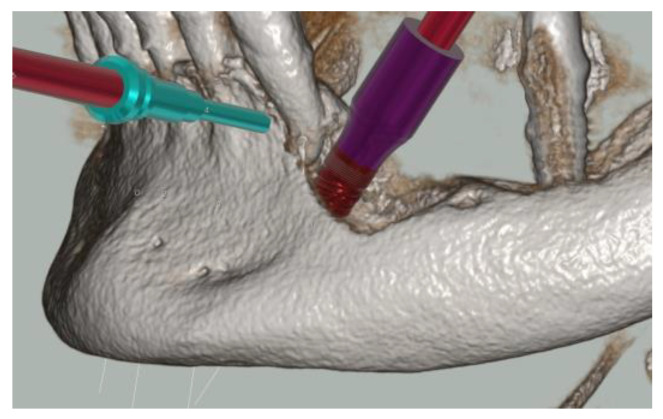
The 3D reconstruction of the bone deficiency in zone 3.4.

**Figure 5 healthcare-12-00332-f005:**
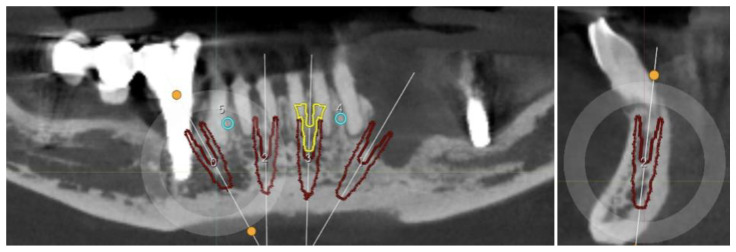
CBCT slice showing the digital planification, details of the 3.75 mm Conic CT implant (red), and the On-Flat Standard abutment (yellow).

**Figure 6 healthcare-12-00332-f006:**
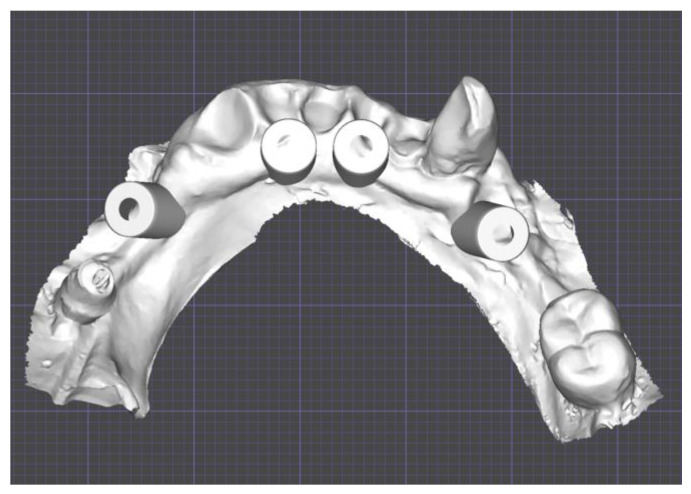
Scan abutments transferred from the surgical planification on Implant3D^®^ to the prosthetic planification on Exocad^®^.

**Figure 7 healthcare-12-00332-f007:**
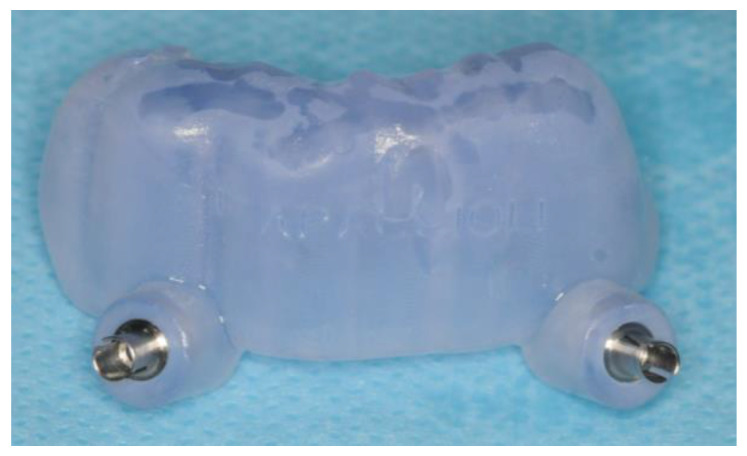
The 3D printed support pin placement guide.

**Figure 8 healthcare-12-00332-f008:**
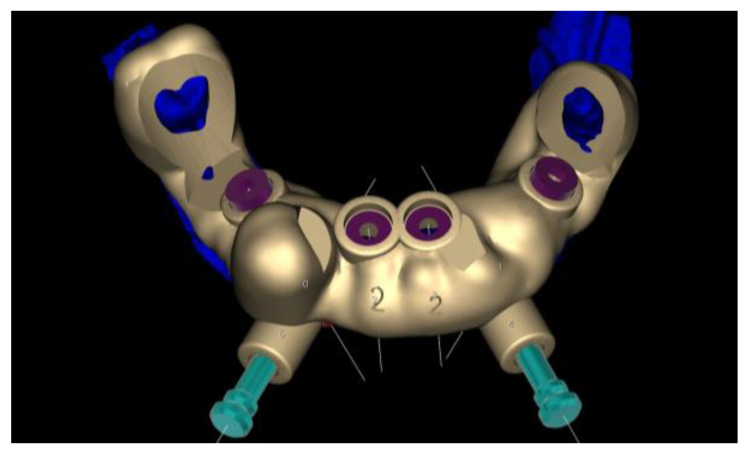
View of the surgical guide.

**Figure 9 healthcare-12-00332-f009:**
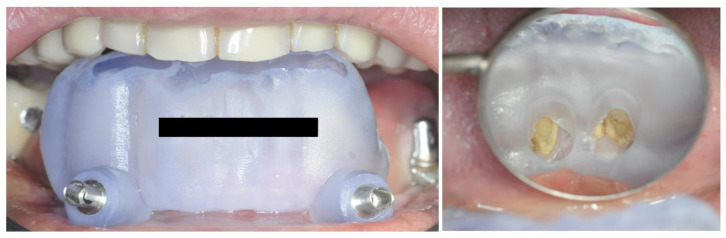
Fit check of the support pin placement guide.

**Figure 10 healthcare-12-00332-f010:**
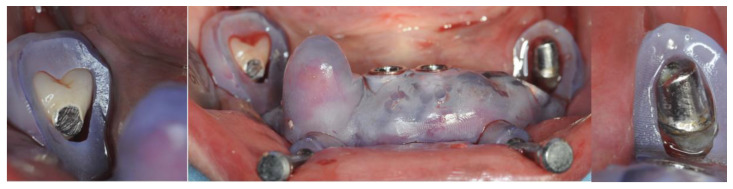
Fit check of the implant placement guide.

**Figure 11 healthcare-12-00332-f011:**
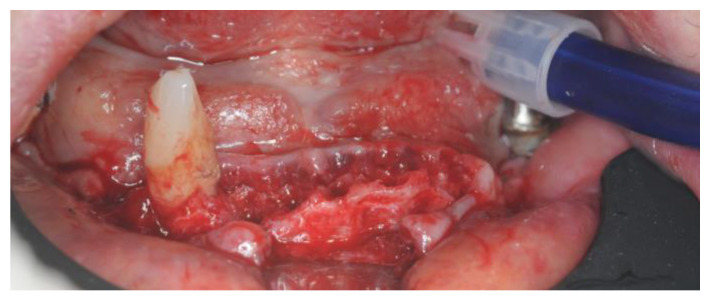
Alveolar ridge after the osteotomy.

**Figure 12 healthcare-12-00332-f012:**
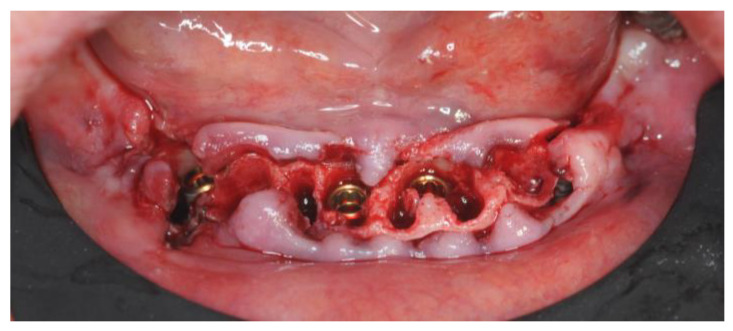
Implant placement.

**Figure 13 healthcare-12-00332-f013:**
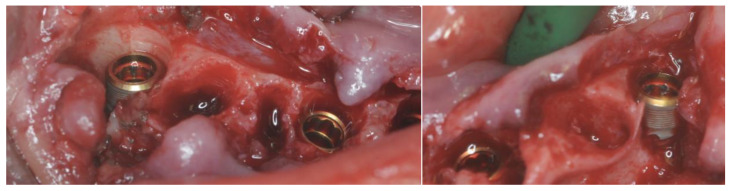
Details of the bone deficiency in zones 4.4 and 3.4.

**Figure 14 healthcare-12-00332-f014:**
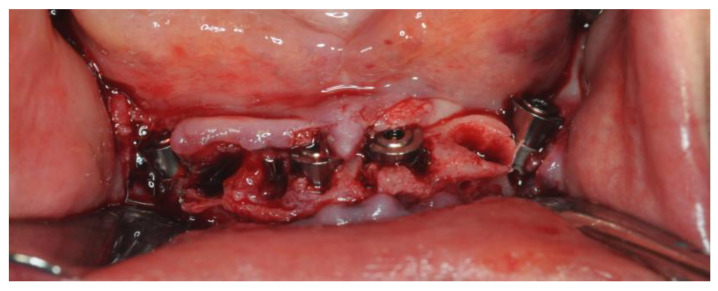
Screwed-in On-Flat^®^ abutments.

**Figure 15 healthcare-12-00332-f015:**
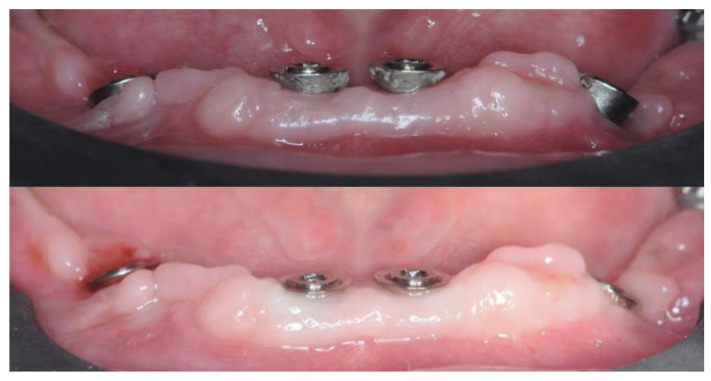
Soft tissue healing and On-Flat^®^ swap to address volume loss.

**Figure 16 healthcare-12-00332-f016:**
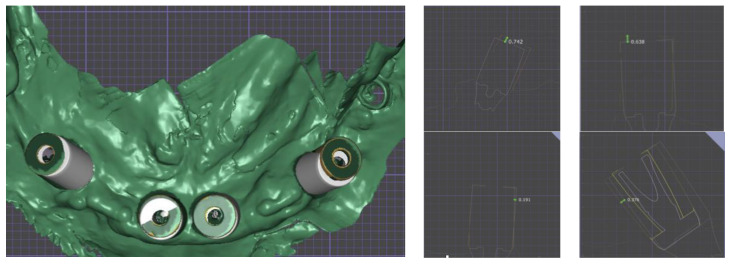
Superimposition of the pre-operative scan abutments’ placements and the post-operative scan. The right panels show the side views of the comparison with measured gaps in millimeters for reference.

**Figure 17 healthcare-12-00332-f017:**
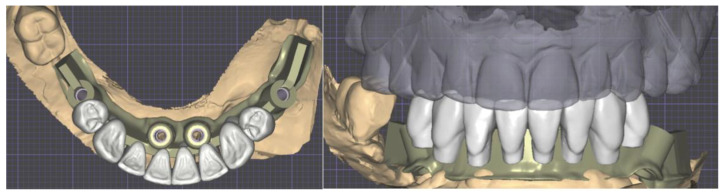
Substructure and dental placement generated with Exocad^®^.

**Figure 18 healthcare-12-00332-f018:**
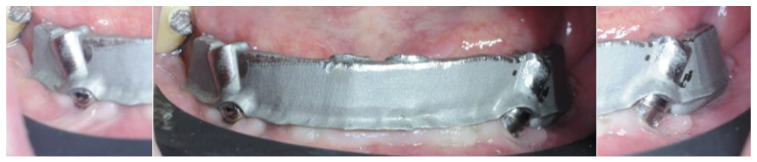
Details of the passivized substructure.

**Figure 19 healthcare-12-00332-f019:**
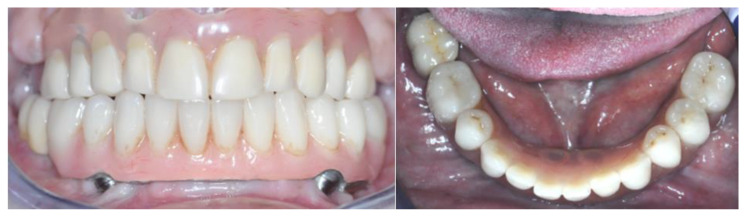
Vestibular and occlusal view of the final prosthesis.

**Figure 20 healthcare-12-00332-f020:**
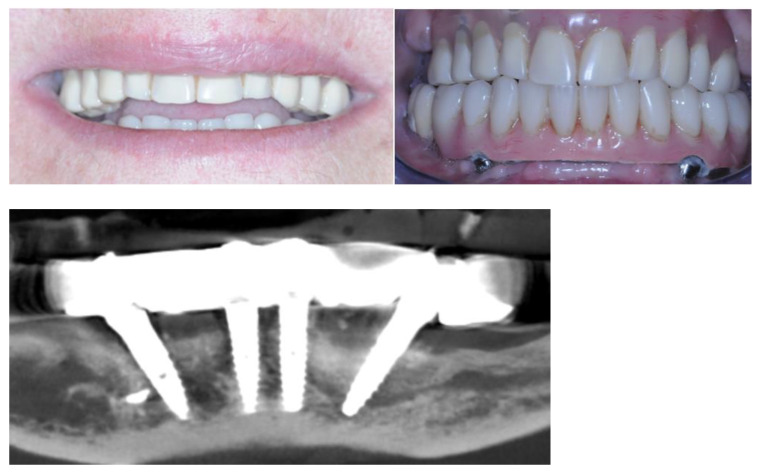
Twelve months post-op.

## Data Availability

Data are contained within the article.

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
