# Peer review of "Lower Jaw Full-Arch Restoration: A Completely Digital Approach to Immediate Load"

_healthcare, 2024, doi:10.3390/healthcare12030332_

Round 1
Reviewer 1 Report
Comments and Suggestions for Authors
The authors have presented a case with a digital approach; however, it may benefit from incorporating more innovative elements. While the fully digital workflow is widely utilized in practices and has been extensively discussed in conferences and meetings over the past years, the case could potentially introduce novel aspects or perspectives.
Additionally, it is noted that the digital planning in this case lacks a prosthetically driven approach, and the lingual placement of the implant to the lower anterior teeth is highlighted in the final result. Furthermore, the absence of a comprehensive restorative space evaluation, a clear case introduction, and a detailed material section are aspects that could enhance the overall presentation.
In the final restoration, there are observed shortcomings, including suboptimal AP spread, an apparent deviation in the occlusal plane, and insufficient cleansing space, among other considerations. Regrettably, the case report fails to address these issues and offer a thorough discussion.
While the presented work falls short as a comprehensive case report, the reviewer suggests a more critical evaluation of whether it meets the criteria for a successful restoration.
Author Response
Dear Reviewer,
Thank you for your feedback. We welcome the opportunity to address your comments and offer additional insights into the various issues you raised.
1. We acknowledge your suggestion and have taken care to emphasize the innovative aspects of our digital approach. While the digital workflow is widely utilized, we believe this clinical case, marked not only by a meticulous analysis of the digital method in each phase but also by its extensive limitations (explained in the responses to other reviewers and in the text), serves as an emblem of how a completely digital design can constitute an additional comprehensive method in resolving complex cases. You can identify the enhancements in the text highlighted in yellow.
2. To assess the potential extension of the cantilever concerning implant positioning, precise millimetric measurements were conducted using the designated design software. Photographs can be deceptive, and in comparison to an AP of 9.5 mm, the cantilever measured 9.8 mm. The prosthesis design underwent careful scrutiny in all aspects to meet the patient's needs and ensure long-term stability of the rehabilitation in alignment with biomechanical criteria.
3. The decision to position the anterior implants more lingually stemmed from a thorough evaluation of available bone and the typical resorption in such conditions. The prosthesis, crafted prior to surgery, was developed based on these considerations. Considering the patient's aversion to GBR procedures, we opted for a more lingual approach when faced with the choice between a lingual emergency and the absence of buccal bone, deeming it the more favorable option.
4. Cleaning of the implant space and level: A one-year follow-up demonstrates effective cleanability. This is primarily attributed to two factors: 1) the metal substructure was fashioned to have a convex and intimate contact with the underlying gum. 2) the implants were positioned at the bone level and only the abutment reaches the tissue level, providing an additional barrier against bacterial infiltration.
5. Regarding the canting of the occlusal plane, we acknowledge the challenge of fully appreciating it from photos without a reference like the Fox plane. Nevertheless, we made it a priority to implement all necessary clinical strategies to prevent a similar issue, given the constraints of not being able to address the rehabilitation of the upper arch.
In conclusion, we value your constructive criticism and demonstrate our commitment to enhancing both clinical practice and the presentation of complex cases. We trust that these responses offer a clearer perspective on the considerations and decisions made throughout the case.
Best regards
Reviewer 2 Report
Comments and Suggestions for Authors
Dear Authors
The manuscript brings an interesting case report, with some innovative technology, but in my opinion needs some improvements before be eligible for publication:
1- Introduction- Please be more specific about what you are introducing, the information is very diffuse on digital technology, and not very concise on the treatment planned and executed; please explore more each step of your treatment plan;
2- Materials and Methods- I missed images of the bone instrumentation using the surgical guide; there is no picture of the temporary prosthesis; no image of the scan bodies used to do the patient scanning; why didn't you replace the upper denture, since it is evident that it was not adequate anymore?
3- Discussion- Please link what was done in the case reported with studies and articles that used the same methods. Readers need to see where and who embraced these methods and supported your treatment plan; there are studies with similar methods published to compare; where, in dentistry, these technologies are been used, its advantages and disadvantages ; it is not clear what was done and how was installed the temporary prosthesis, for an example;
Author Response
Dear Reviewer,
we appreciate your constructive feedback on our manuscript. We have carefully considered your insightful comments and have made the necessary revisions to improve the overall quality of the case report.
- The introduction has been revised to be more concise and focused. The information is now more specific, providing a clearer understanding of the treatment planned and executed. Each step of our approach has been explored in greater detail, offering readers a more in-depth view of the innovative technology applied in the reported case.
- Materials and Methods: We want to clarify that we adhered to the permitted limits for the number of photos, and a decision was made not to include images of certain phases due to their lower resolution. However, readers can observe the use of scan bodies in Image No. 6 and No. 16, providing visual insight into our methodology.
- Text Integration: The specific requests regarding therapeutic choices for the upper arch and the clinical management of the provisional prosthesis have been integrated into the text. These additions aim to provide a more comprehensive understanding of our decision-making process.
- Discussion: In response to your suggestion, relevant studies with a similar approach have been cited in the discussion section. This emphasizes the validation of our chosen method and provides readers with comparative insights into the broader literature.
We believe these revisions significantly enhance the manuscript and align it more closely with the standards for publication. Your guidance has been invaluable, and we look forward to your consideration of the improved version.
Best Regards
Reviewer 3 Report
Comments and Suggestions for Authors
The article describe in a correct manner a guided surgical protocol without flapless approach. I appreciated a lot the final bone crest recontour and the subcrestal insertion of the implants.
Please in the discussion underline this important step.
Author Response
Dear Reviewer,
Thank you for your valuable feedback. We are pleased that you found the guided surgical protocol and the specific steps highlighted in the article noteworthy. The final bone crest recontour and the subcrestal insertion of implants are indeed crucial aspects of the procedure, playing a pivotal role in achieving optimal aesthetic and functional outcomes in guided surgical procedures. These steps are essential for ensuring proper support and stability for the implants, thereby contributing to long-term success. Adapting the protocol to each clinical case is essential to address any challenges that may arise. Factors such as bone density, soft tissue health, and the patient's overall oral health status should be carefully considered to tailor the surgical approach accordingly.
Please note that the improvements you suggested have been incorporated into the text and are highlighted in yellow for your convenience.
Best Regards
Reviewer 4 Report
Comments and Suggestions for Authors
Dear Author,
The discussion and introduction sections are excessively lengthy. Kindly simplify it by introducing the primary concept that you intend to highlight.
Line 103… Lower maxilla?? Please correct the statement
Kindly provide higher quality images for Figures 2 and 20. (beam hardening artifact ??)
What is the radiopaque image in the area between 4.3-4.4 in Fig. 20?
Please justify the administration of 2 g of Amoxicillin + Clavulanic acid prior to surgery. Should prophylaxis be administered to this patient?
The practitioner and patient should consider possible clinical circumstances that may suggest the presence of a significant medical risk in providing dental care without antibiotic prophylaxis, as well as the known risks of frequent or widespread antibiotic use. For infective endocarditis prophylaxis, American Heart Association guidelines (updated with a scientific statement;in 2021) support premedication for a relatively small subset of patients. This is based on a review of scientific evidence, which showed that the risk of adverse reactions to antibiotics generally outweigh the benefits of prophylaxis for many patients who would have been considered eligible for prophylaxis in previous versions of the guidelines. Concern about the development of drug-resistant bacteria also is a factor.
Author Response
Dear Reviewer,
thank you for your feedback. We have diligently addressed your comments as outlined below:
- Introduction and Discussion Sections: The introduction and discussion sections have been thoroughly revised and condensed, aligning with your guidance to emphasize the primary concept.
- The error at line 103 has been rectified.
- Images in Figures 2 and 20 have been enlarged while adhering to text formatting limits.
- The radiopaque image in the area between 4.3-4.4 in Figure 20 represents an osteointegrated residue from the previous implant. This specific CBCT cut was chosen for its optimal representation of the overall postoperative scenario.
- Justification for Antibiotic Administration: despite recent guidelines suggesting the patient may not necessarily require antibiotic prophylaxis, we have opted for a more cautious approach. This decision is grounded in the prolonged flap detachment exceeding 2 hours, bone remodeling, gingival repositioning, and other intricate surgical phases. We have adhered to pre-existing literature protocols predating the latest updates. Here a DOI of an article that for example we referred to (DOI: 10.1563/1548-1336-49.1.93).
Best Regards
Round 2
Reviewer 2 Report
Comments and Suggestions for Authors
Dear Authors
Thank you for improving the manuscript .
Best regards
Reviewer 4 Report
Comments and Suggestions for Authors
Dear Author,
Thank you for your corrections. The manuscript is acceptable as it is.